# Current Evidence for *Corynebacterium* on the Ocular Surface

**DOI:** 10.3390/microorganisms9020254

**Published:** 2021-01-27

**Authors:** Takanori Aoki, Koji Kitazawa, Hideto Deguchi, Chie Sotozono

**Affiliations:** Department of Ophthalmology, Kyoto Prefectural University of Medicine, Kyoto 602-0841, Japan; taka526@koto.kpu-m.ac.jp (T.A.); hdeg1988@koto.kpu-m.ac.jp (H.D.); csotozon@koto.kpu-m.ac.jp (C.S.)

**Keywords:** *Corynebacterium*, infectious keratitis, conjunctivitis, resistant bacteria, fluoroquinolone, *C. macginleyi*, *C. propinquum*, *C. mastitidis*

## Abstract

*Corynebacterium* species are commonly found in the conjunctiva of healthy adults and are recognized as non-pathogenic bacteria. In recent years, however, *Corynebacterium* species have been reported to be potentially pathogenic in various tissues. We investigated *Corynebacterium* species on the ocular surface and reviewed various species of *Corynebacterium* in terms of their antimicrobial susceptibility and the underlying molecular resistance mechanisms. We identified a risk for *Corynebacterium*-related ocular infections in patients with poor immunity, such as patients with diabetes or long-term users of topical steroids, and in those with corneal epithelial damage due to trauma, contact lens wear, lagophthalmos, and trichiasis. The predominant strain in the conjunctiva was *C. macginleyi*, and the species associated with keratitis and conjunctivitis were *C. macginleyi*, *C. propinquum*, *C. mastitidis*, *C. pseudodiphtheriticum*, *C. accolens, C. striatum*, *C. xerosis*, and *C. bovis*. Overall, *Corynebacterium* species present on the ocular surface were resistant to quinolones, whereas those in the nasal cavity were more susceptible. The prevalence of fluoroquinolone-resistant *Corynebacterium* has not changed in the past 10 years; however, *Corynebacterium* species remain susceptible to third-generation cephems. In conclusion, the use of third-generation cephems should be a reasonable and pragmatic approach for treatment of ocular infections caused by *Corynebacterium* species.

## 1. Introduction

*Corynebacterium* species are frequent constituents of the normal bacterial flora of the conjunctival sac and are common as well on human skin and mucous membranes and in the intestines [1,2]. One important species, *Corynebacterium diphtheri*, is the pathogenic bacterium that causes diphtheria, an upper respiratory infection generally characterized by sore throat and high temperature. In severe diphtheria cases, respiratory obstruction due to the inflammation of the tonsils, oropharynx, and pharynx can lead to death as the worst-case scenario. However, in general, *Corynebacterium* species seem to have low pathogenicity; therefore, the isolation of *Corynebacterium* species from infected tissue has been considered to arise as a result of mishandling or contamination [3,4].

*Corynebacterium* species can be found commonly on the ocular surface, in conjunction with other indigenous bacteria, such as *Staphylococcus epidermidis* and *Propionibacterium acnes* [1,5]. These various commensal bacteria aid in preventing ocular surface from invasion by foreign organisms [6]. However, in immunocompromised patients, many recent reports have demonstrated that *Corynebacterium* species can be potentially pathogenic when present on the ocular surface [7,8,9,10], as this infection has been associated with cases of endocarditis of the aortic and mitral valves [11], granulomatous mastitis [12], and pelvic osteomyelitis [13]. Many reports have been based on the case series [7,14,15,16,17,18,19,20,21]. Therefore, the frequency of occurrence of various *Corynebacterium* species on the ocular surface compared to other organs is not well characterized, and the mechanism by which *Corynebacterium* species function as pathogenic organisms is also unclear. Moreover, the antimicrobial susceptibility pattern in *Corynebacterium* species, as with other bacteria like *Staphylococcus aureus*, has been changing in recent years [22]. Consequently, a full understanding of the pathogenicity of *Corynebacterium* species awaits a systematic review of the various ocular infection cases possibly caused by *Corynebacterium* species.

The purpose of the present review is to investigate the *Corynebacterium* species occurring on the ocular surface and to review the antimicrobial susceptibility of these species from the perspective of the underlying molecular resistance mechanism.

## 2. Corynebacterium Species

The genus *Corynebacterium* consists of 137 species, and 10 of these have been isolated from corneal infections (Table 1). *Corynebacterium* species are commonly found in the conjunctiva of healthy adults and have been recognized as non-pathogenic commensal bacteria on the ocular surface [7,8]. One study of 990 patients prior to cataract surgery revealed positive culture results for *Corynebacterium* species in the conjunctival sac in 46.4% (460/990), which was much higher than the results for methicillin-sensitive coagulase-negative staphylococci (MS-CNS), at 22.5% (223/990), or methicillin-sensitive *Staphylococcus aureus* (MSSA), at 4.4% (44/990) [5].

Hoshi and associates evaluated 183 strains of *Corynebacterium* species in patients during their preoperative examinations prior to cataract surgery. Positive culture results revealed that *C. macginleyi* was the predominant strain (84%) in the conjunctiva, followed by *C. accolens* at 7%, *C. propinquum* at 3%, *C. amycolatum* at 2%, *C. jeikeium* at 2%, and *C. mastitidis* at 2% [23]. The presence of *Corynebacterium* species occurs in a tissue-specific manner; for instance, *C. macginleyi* is a dominant strain in the conjunctiva and was first described by Riegel and associates as a strain isolated from eye samples [24]. By contrast, *C. accolens* and *C. propinquum* are two major species in the nasal cavity, at 44% and 31%, respectively, whereas *C. macginleyi* accounts for only 3% of the isolates from nasal samples [23].

*Corynebacterium* species is thought to be a possible pathogen responsible for keratitis in some cases, such as biofilm formation in immunosuppressed persons. In fact, cases of *Corynebacterium*-associated ocular infection, including conjunctivitis, bacteria keratitis, glaucoma bleb-related infections, and endophthalmitis, have been reported [7,8,14,15,16,17,18,19,20,21,25,26,27,28,29,30,31], have led to suspicion of involvement of *Corynebacterium* species in immunocompromised cases. Many case reports have implicated *C. macginleyi* as a main strain that causes conjunctivitis and bacteria keratitis [7,8,17,18,19,20,28,32].

In this review, we have listed the cases of ocular surface infection, including keratitis, corneal ulcer, and conjunctivitis, caused by *Corynebacterium* species (Table 1). The identified species were *C. macginleyi* (*n* = 24), *C. propinquum* (*n* = 2), *C. pseudodiphtheriticum* (*n* = 2), *C. striatum* (*n* = 2), *C. xerosis* (*n* = 1), and *C. mastitidis* (*n* = 1) [7,8,14,15,16,17,18,19,20,21,25,26,27]. Most of the infected patients had risk factors, such as diabetes or long-term use of steroid treatments after corneal transplantation [14,16,19,25] that led to poor immunity, or they had experienced corneal epithelial damage due to trauma [20], contact lens wear [7,16,26,27], lagophthalmos, or trichiasis [21]. Interestingly, mild infectious cases, like conjunctivitis, were more often observed in cases of *C. macginleyi* infection, whereas other *Corynebacterium* strains seem to cause more severe infectious diseases [14,16,21]. In fact, many severe keratitis cases reported from India were caused by strains identified as *C. propinquum* (*n* = 3), *C. pseudodiphtheriticum* (*n* = 1), *C. striatum* (*n* = 1), and *C. bovis* (*n* = 1) [9], suggesting that the causative agents could be either indigenous bacteria or foreign strains.

### 2.1. C. macginleyi

Riegel and associates reported lipophilic coryneform bacteria from eye specimens that they named *C. macginleyi* [24]. The presence of this species in the normal conjunctiva is not surprising because the meibomian glands produce meibum, an oily substance [24]. In fact, *C. macginleyi* is the most commonly isolated strain in the conjunctiva, and it is also recognized as the most common causative agent of opportunistic ocular infections.

The rDNA sequence of *C. macginleyi* shows a 98.7% similarity to that of *C. accolens*. Eguchi and associates showed that *C. macginleyi* was highly resistant to fluoroquinolones, with 12 of 16 tested isolates showing resistance. They attributed this resistance to overuse of fluoroquinolones eye drops in the ophthalmology field. *C. macginleyi* is an opportunistic pathogen that causes several ocular infections, including conjunctivitis [7,8,18], corneal keratitis [17,19], glaucoma bleb-related infections [30,31], and endophthalmitis [10,29]. *C. macginleyi* can also cause non-ocular infections, such as bladder catheter infections [33], intravenous catheter infections [34,35], and septicemia [36].

### 2.2. C. accolens

*C. accolens* (previously CDC group G-1) was first described from human clinical specimens, such as wound drainage, endocervix samples, sputum, and throat swab specimens collected over a 30-year period [3,37]. Hoshi and associates revealed that *C. accolens* was the second most frequent bacterial species, accounting for 7% of the bacteria in the conjunctiva and 44% in the nasal cavity in the healthy volunteers [23]. One case with bacterial conjunctivitis had *C. accolens* identified in the conjunctival sac [8]. However, evidence confirming the *Corynebacterium* isolates as the causative agents of the infection was not provided.

### 2.3. C. propinquum

*C. propinquum* was proposed as the CDC coryneform ANF-3 bacterium by Riegel and associates [38]. Two cases of keratitis caused by *C. propinquum* have been reported. One case was a 94-year-old woman who had undergone corneal transplantation for Fuchs corneal dystrophy, and she had used long-term 0.1% fluorometholone eye drops as prophylaxis against graft rejection. Corneal infection had occurred at the loosened suture region of the graft. After medication with 5% cefazolin eye drops and 1% gentamicin eye drop hourly and 0.1% fluorometholone eye drops twice daily, her visual acuity improved to 20/160 [14]. Another case was a 44-year-old woman with a persistent corneal epithelial defect. She had a past history of type 1 diabetes, proliferative diabetic retinopathy, and hemodialysis due to diabetic nephropathy. The ocular infection was attributed to a persistent corneal epithelial defect due to neurotrophic keratopathy, possibly caused by poor diabetes control. The patient was treated with gatifloxacin and cefmenoxime eye drops six times daily and complete eye closure with an eye patch to facilitate reepithelialization. Her final visual acuity recovered to 0.02 (20/1000) [16].

*C. propinquum* is typically commensal on the human nasopharynx and skin [23,39]. *C. propinquum* has been implicated in various opportunistic infections, such as pulmonary infection [40] and infective endocarditis [41]. The majority of *C. propinquum* strains are constitutively resistant to macrolide drugs [23,42], and both of the above cases were resistant to erythromycin [23].

### 2.4. C. amycolatum

*C. amycolatum* is a non-lipophilic and fermentative *Corynebacterium* species, first described by Collins and associates from swabs of the skin of healthy people [43]. One case report has appeared of orbital implant infection after eye evisceration caused by *C. amycolatum* [44], but no reports of conjunctivitis or keratitis have been published. *C. amycolatum* is a normal inhabitant of skin and mucous membranes and promotes the epidermal growth factor receptor-dependent induction of the antimicrobial protein RNase7. The relationship between *C. amycolatum* and RNase7 may control the growth of *Corynebacterium* species on human skin [45].

### 2.5. C. jeikeium

The species *C. jeikeium* was isolated from bacterial endocarditis following cardiac surgery [46]. *C. jeikeium* is part of the normal flora of the skin, especially in inpatients. Many studies have reported multi-resistance to antibiotics in *C. jeikeium* strains [47,48,49].

### 2.6. C. mastitidis, C. lowii, and C. oculi

*C. mastitidis* was first found in sheep with mastitis [50] and this species can stably colonize the ocular mucosa, where it provides a related beneficial local immunity [51]. Bernard and associates analyzed *C. mastitidis* recognized by Eguchi in Japan and Vandamm in Belgium and Switzerland [8,52], and described *C. lowii* and *C. oculi* as two new species, separate from *C. mastitidis*. *C. mastitidis* was detected in contact-related keratitis in Japan; however, unlike *C. macginleyi, C. mastitidis* was very sensitive to both levofloxacin and ciprofloxacin [27].

## 3. Laboratory Examinations

### 3.1. The Ocular Manifestations of Corynebacterium Species

*Corynebacterium*-associated conjunctivitis is characterized by variable symptoms, including hyperemia, foreign body sensation, discharge, and a burning sensation [8,18,28,32]. Bacterial keratitis caused by *Corynebacterium* species shows clinical features by slit lamp microscopy that range from mild cases with a small infiltration on the surface of cornea to severe cases with corneal ulcerations with round to oval shapes at the center of cornea, but the ocular infections show no distinctive findings [9,17,19,25,53,54]. This infection is more frequently found in patients undergoing immunosuppressive therapy after corneal transplantation. As shown in Table 1, age differences do not seem to affect the ocular infection by *Corynebacterium* species. Trauma, contact lens wear, and corneal damage due to trichiasis and severe dry eye can also exacerbate the ocular infection [7,15,20,21,26,27]. However, *Corynebacterium* species isolated as part of the normal flora in the conjunctival sac showed a significant association with age, male sex, and glaucoma eye drop use in a multivariate analysis [5], suggesting that clinical information would be helpful for an accurate diagnosis.

Figure 1 shows the findings for an 89-year-old man who had undergone corneal transplantation for lattice corneal dystrophy eight years previously. He had applied 0.1% fluorometholone eye drops twice daily for the prevention of allograft rejection, accompanied by moxifloxacin eye drops for post microbial keratitis as prophylaxis. At his regular visit, corneal infiltration was found at the suture site. A microbial examination identified *Corynebacterium* species that were resistant to levofloxacin, erythromycin, cefmenoxime, and fosfomycin and were susceptible to arbekacin, vancomycin, and chloramphenicol. The patient was treated with 0.3% chloramphenicol eye drops, 1% vancomycin ointment, and 1% cefmenoxime eye drops, and the keratitis was resolved.

### 3.2. Microscopy Examinations

Microscopy examinations of the discharge/fluid from the infected conjunctival sac and of corneal scrapings are helpful for identifying the causative bacteria. *Corynebacterium* species are gram-positive, rod-shaped, non-branching, non-motile, catalase positive, and oxidase negative bacteria. They grow in aerobic conditions, and *Corynebacterium* species are widely present in nature, in water, soil, and plants. The range in size from 0.3–0.8 μm in diameter and 1–8 μm in length. They look like the letters “I, N, T, V, W, or Y” at 1000× magnification, and show the apical growth typical of a bacillus, often exhibiting a club-shaped morphology at one or both ends. *Corynebacterium* species are commonly isolated as indigenous bacteria from the normal ocular flora as well as the mucosa and skin. However, as shown in Figure 2, *Corynebacterium* species are phagocytosed by polymorphonuclear leucocyte, indicating that they can have a potential impact on infection.

### 3.3. Culture Tests

Most *Corynebacterium* species can be isolated from a 5% sheep blood agar, and they often form staphylococcal-like colonies. The growth of lipophilic *Corynebacterium* species is enhanced by adding 0.1% Tween 80 to the medium, while blood agar under aerobic conditions helps the growth of non-lipophilic *Corynebacterium* species. C. *macginleyi* is a lipophilic *Corynebacterium* and requires lipid for growth. Clinical laboratories typically report these organisms as “*Corynebacterium* spp.” based on visualization and catalase reactions only, which could lead to misdiagnosis. When other commensal bacteria from the respiratory system material or urinary system material are present, identification of *Corynebacterium* species may not always be necessary, as they are unlikely to be the causative pathogens of an infection. Therefore, *Corynebacterium* should be detected in clinical samples from normally sterile sites. However, the normal bacterial flora on the ocular surface includes *Corynebacterium* species [1], suggesting that the identification of *Corynebacterium* species should be performed with clinical findings in cases when immunity is severely affected, as those patients could develop infections caused by *Corynebacterium* species.

### 3.4. Antimicrobial Susceptibility Testing

Antimicrobial susceptibility testing is a method for determining possible drug resistance and for assuring the susceptibility to the drugs of choice for a particular bacterial species. Antimicrobial susceptibility testing should be performed for cases highly suspicious for *Corynebacterium* being a causative pathogen based on smear speculation or isolates from sterile materials. The minimal inhibitory concentrations (MIC values) of a drug for lipophilic or non-lipophilic *Corynebacterium* are typically determined using 5% lysed horse blood added to agar and adjusted with Ca and Mg with microfluid dilution. The bacteria are incubated for 24–48 h at 35 °C, with 48 h needed for lipophilic *Corynebacterium* species. Growth on control agar is compared to growth on the drug-containing agar to determine susceptibility or resistance. The disk diffusion method is a standardized technique for testing rapidly growing pathogens. However, many reports suggest some limitations to the use of the disc diffusion method [55,56], as some cases do not accurately display a visual resistance reaction.

## 4. The Susceptibility of *Corynebacterium* to Antibiotics

The susceptibility of *Corynebacterium* to drugs varies depending on the species, according to published reports [7,8,9,10,14,16,18,19,21,22,23,27,28,32,53,54,57,58,59,60]. In the ophthalmology field, large clinical cohorts of *Corynebacterium* on the ocular surface have not yet been sufficiently studied. Table 2 shows the previous clinical research where *Corynebacterium* species were identified with conjunctival swabs in cases with infectious ocular surface diseases or at the time of preoperative examination for cataract surgery. *C. macginleyi* accounted for a large proportion of the ocular infection [7,8,16,18,19,23,28]. In 1998, Funke and associates were the first to report *C. macginleyi* in 15 cases of conjunctivitis or corneal ulcer [7]. They also found that *C. macginleyi* was susceptible to penicillin and ceftriaxone as well as to ofloxacin or ciprofloxacin, which are quinolone antibiotics [7]. However, in recent years, many cases of resistance to quinolones have been reported for *C. macginleyi*. For example, Eguchi and associates reported that 20 cases with *Corynebacterium* strains identified on the ocular surface were susceptible to erythromycin at 45% or to levofloxacin or ciprofloxacin at 25%, whereas the susceptibility to cefmenoxime was 100% [32]. Another report for samples obtained from the conjunctival sac in patients prior to cataract surgery revealed that 51 isolates of *C. macginleyi* were highly susceptible to both penicillin and erythromycin, but showed low susceptibility to levofloxacin at 64% [23]. We previously also reported one case with infectious keratitis following phototherapeutic keratectomy, caused by a fluoroquinolone-resistant *Corynebacterium* species [53]. We also examined the *Corynebacterium* species isolated from an ocular infection that included the conjunctiva and found that 54% of the *Corynebacterium* species were resistant to levofloxacin [22]. Many levofloxacin-resistant *Corynebacterium* strains have been reported in Japan, suggesting the possibility of overuse of quinolones eye drops in Japan. However, as shown in Table 2, *Corynebacterium* species reported in India have also revealed low susceptibility to ciprofloxacin, at 50–62% [9,59]. Other reports from western countries, including Australia, the USA, Ireland, Germany, Canada, and Italy, have shown a higher susceptibility to fluoroquinolone in *Corynebacterium* species [14,15,17,18,20,28,57,60], suggesting that strains of *Corynebacterium* can be responsible for susceptibility differences, rather than the use of topical antibiotics. Antimicrobial susceptibility may also be affected by strain types. *C. propinquum* has a known resistance to macrolides based on studies in non-ocular tissues [23,42]. Case reports from Australia and Japan have also indicated high resistance to erythromycin [14,16]. Hoshi and associates have reported that the susceptibility to erythromycin in *C. propinquum* isolated from the conjunctiva and nose was 74%, while *C. macginleyi* was highly susceptible [23]. In the clinical setting, the strain of *Corynebacterium* may not be identified in most cases, but *Corynebacterium* overall show mild to high resistance to quinolone antibiotics, whereas *Corynebacterium* on the ocular surface are susceptible to third-generation cephems. Therefore, the possibility of strain-dependent antimicrobial susceptibility should be kept in mind.

We previously examined the trend of resistance to antibiotics in ocular infections by *Staphylococcus aureus*, coagulase-negative staphylococci, and *Corynebacterium*. The prevalence of methicillin-resistant *Staphylococcus aureus* (MRSA) and methicillin-resistant coagulase-negative staphylococci (MR-CNS) has decreased from 52% to 22% and from 47% to 25%, respectively, over a 10-year period, whereas 54% of the *Corynebacterium* species still remain resistant to levofloxacin, and the prevalence of resistant *Corynebacterium* species has not changed [22]. Levofloxacin-resistant *Corynebacterium* species are highly resistant (28%) to erythromycin, as well as to levofloxacin, while they are fully susceptible to cefmenoxime (100%) [22]. We also investigated 264 cases diagnosed as ocular surface infection at the Baptist Eye Institute in Kyoto, Japan, between April 2014 and March 2016, using bacterial cultures from the conjunctival sac and the nasal cavity. We isolated 77 strains of *Corynebacterium* from the conjunctival sac, and 176 strains were detected in the nasal cavity. The susceptibility to erythromycin was 40% and to levofloxacin was 45%, whereas cefmenoxime sensitivity was 96% (Table 3), in agreement with the findings of Eguchi and associates [32]. Conversely, the susceptibilities to erythromycin, levofloxacin, and cefmenoxime in the *Corynebacterium* species from the nasal cavity were 69%, 74%, and 94%, respectively. One possible explanation for this difference in susceptibility between strains from the conjunctival sac and the nasal cavity is that the conjunctival sac was more exposed to antibiotics, such as levofloxacin, compared to the nasal cavity.

Eguchi and associates performed multilocus sequence typing (MLST) to classify the lineage of *C. macginleyi*, and they found that 13 of 16 isolates were clustered into the same group. In addition, all the isolates were resistant to fluoroquinolone, indicating that fluoroquinolone resistance was widely prevalent in the *C. macginleyi* species on the ocular surface. However, whether the *Corynebacterium* isolates recovered from specimens were truly the causative agents of the infection is not clear. Nevertheless, these findings suggest that overuse of antibiotics can affect the prevalence of drug-resistant bacteria.

The susceptibility of *Corynebacterium* species to fluoroquinolones varies among different countries. A low susceptibility to fluoroquinolones has been reported in Japan, while reports in Switzerland and Canada show high susceptibility to these drugs [7,18]. This may reflect the fact that fluoroquinolones are often the first choice for the treatment of conjunctivitis in Japan. Therefore, bacteria resistant to fluoroquinolones are more likely to be reported, whereas the susceptibility to third-generation cephems still remains at a high level (Table 1). Multidrug-resistant *Corynebacterium* are becoming a larger issue in ophthalmology [61,62], so third-generation antibiotics would be recommended as the first choice when a *Corynebacterium* infection is suspected.

The difference in drug susceptibility in different tissues may be attributed to the types of *Corynebacterium* present. In systemic studies, *C. striatum* is the most frequent *Corynebacterium* identified from blood and is the most common cause of bacteremia. *C. striatum* is frequently resistant to penicillin, cephems, and fluoroquinolone and is often treated with vancomycin. However, *C. striatum*-related keratitis, which was first reported in 1989 and 1991 [25,26], is highly susceptible to penicillin. Neemuchwala and associates evaluated the antimicrobial susceptibility pattern of 1970 strains of *Corynebacterium* identified between 2011 and 2016, and they found that 931 (47%) strains of *C. striatum*, 190 (10%) strains of *C. amycolatum*, and 216 strains (11%) of *C. peudodiphtheritium/C. propinquum* showed low susceptibility to penicillin (15%), followed by erythromycin (15%), and ciprofloxacin (27%) [42], suggesting that these strains are less sensitive to fluoroquinolones as well. Consequently, fluoroquinolone-resistant *Corynebacterium* should be considered in other fields.

## 5. Genetic Mutations and Drug Resistance

The antimicrobial susceptibility to drug in bacteria has been changing over time, and antimicrobial resistance (AMR) is now an issue everywhere in the world [63]. The World Health Assembly set up a global action plan on AMR in 2015 to deal with the growing problem of resistance to antibiotics. Drug-resistance in pathogens arise though several biochemical mechanisms and is acquired by genetic mutations of the active points of the gene. Gene mutations on the pathogen side render the drug ineffective and prevents the pathogen from being a target of the drug. This mechanism is commonly found in microorganisms such as MRSA. Another drug resistance mechanism in bacteria is chemical modification or breakdown or inactivation of the target drug by enzymes. For instance, penicillin-resistant *Staphylococcus aureus* (other than MRSA) exhibits drug resistance by producing penicillinase and β-lactamase enzymes that break down penicillin. Excretion of the drug from the bacterial cells can also lead to resistance to the drug. For example, *Escherichia coli* and *Pseudomonas aeruginosa* have the RND-type multidrug efflux pumps that can accelerate pumping of drugs out through the AcrAB-TolC system in *E.coli* or the MexAB-OprM and MexXY-OprM systems in *Pseudomonas aeruginosa*, thereby reducing the drug concentration in the cells [64,65,66].

*Corynebacterium* species can acquire drug resistance by genetic mutations of the drug target site. As shown in Table 2, *Corynebacterium* species have low susceptibility to quinolones. Bacteria generally become resistant to quinolones due to an overexpression of efflux pump gene(s) and some gene mutations of the target molecules [67]. The quinolones are broad-spectrum antibiotics with a 4-quinolone skeleton. The quinolones interfere with bacterial growth by inhibiting DNA gyrase (*gyrA*) and topoisomerase IV (*purC*) [68], which are essential bacterial type II topoisomerase enzymes that mediate the winding and unwinding of the DNA and DNA double helix breakage during DNA replication. The quinolones therefore stop DNA replication in the bacteria, resulting in cell death. *Corynebacterium* species lack topoisomerase IV, so DNA gyrase plays a major role in the resistance to quinolones in *Corynebacterium* species [69]. In many quinolone-resistant bacteria, point mutations occur in the quinolone resistance determining region (QRDR) that encodes DNA gyrase and topoisomerase IV. These mutations are attributed to the acquisition of fluoroquinolone resistance [67,70], and 83(Ser) and 87(Asp) are the two most commonly mutated amino acids in quinolone-resistant mutants [71].

For *Corynebacterium* species, a single amino acid mutation in DNA gyrase has occurred most commonly at the QRDRs of gyrA (residues at amino acid positions 83 and 87 to 91) [8,72,73,74]. The loci of the mutations depend on the species. Eguchi and associates reported that point mutations at amino acid positions 83(Ser) and 87(Asp) of the QRDRs were strongly associated with quinolone resistance in *C. macginleyi* [8]. In *C. striatum*, point mutations occur at amino acid positions 87(Ser) and 91(Ala), and the double mutations show an increase in resistance to moxifloxacin [72,73]. For *C. jeikeium* and *C. urealyticum*, as well as in other *Corynebacterium* species, the important amino acid changes occur at positions 87 and 91. First, a point mutation at either amino acid position 83(Ser) and 87(Asp) occurs, leading to a change in the subsequent intrinsic sensitivity to the antibiotics [74], which eventually exacerbates to susceptibility to antibiotics.

Recently, many reports have shown the presence of multidrug-resistant *Corynebacterium* species in systemic diseases, such as sepsis, endocarditis, and granulomatous mastitis [48,75,76,77,78]. At present, six cases of ocular surface infections caused by multi-drug resistant *Corynebacterium* species have been reported [61,62]. All these cases occurred in patients with poorly controlled diabetes, long-term steroid treatment, bullous keratopathy, or Stevens-Johnson syndrome or who used punctal plugs due to severe dry eye disease. The *Corynebacterium* species were not identified; however, the following genes coding for antibiotic resistance, including quinolones, were found: *ermX* (macrolides and lincosamides), *aphA* (aminoglycosides), and *cmx* (chloramphenicol) [47,79,80]. The overuse of antibiotics can drive spontaneous mutation in *Corynebacterium* species [72,73], thereby causing a prevalence of multidrug-resistant *Corynebacterium* species.

Fourth generation quinolone eye drops are now a common choice for prophylactic administration in patients undergoing intraocular surgery [81]. New ophthalmic eye drops containing hexamidine diisethionate have also been developed for the treatment of keratitis, conjunctivitis, and blepharitis [82,83]. Interestingly, the long-term use of glaucoma eye drops has also been reported to change the commensal bacteria flora on the ocular surface [84], although this effect may be a consequence of the addition of benzalkonium chloride (BAC) preservative to the glaucoma eye drops. As antimicrobial treatments develop, following the changes in the commensal bacterial flora in the ocular surface will also be necessary.

## 6. Conclusions

This review confirms that *Corynebacterium* species can work as a causative pathogen in immunocompromised patients. Generally, *Corynebacterium* species are commensal bacteria, and they may be dismissed as a nonpathogenic organism in most cases. However, a suspicion of *Corynebacterium* infection following microscopy examination with Gram staining confirmed that *Corynebacterium* species were phagocytosed by polymorphonuclear leucocytes. In urgent cases, empirical therapy with third-generation cephems, such as cefmenoxime, is a reasonable and pragmatic approach for immediate treatment of the infection. Regardless, switching to a definitive therapy based on the subsequent results of antimicrobial susceptibility testing is critical later on. In fact, quinolone-resistant and multidrug-resistant *Corynebacterium* species have been reported, so the use of clinically and microbiologically appropriate drugs is essential to prevent these bacteria from becoming drug resistant.

## Figures and Tables

**Figure 1 microorganisms-09-00254-f001:**
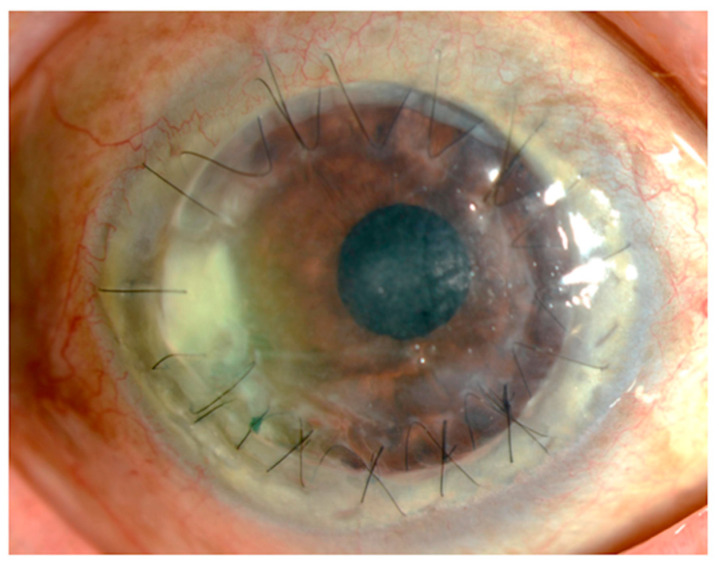
Slit lamp photograph. An 89-year-old man who underwent a penetrating keratoplasty at his age of 81. The corneal infiltration was observed around the graft suture, accompanied with moderate hyperemia.

**Figure 2 microorganisms-09-00254-f002:**
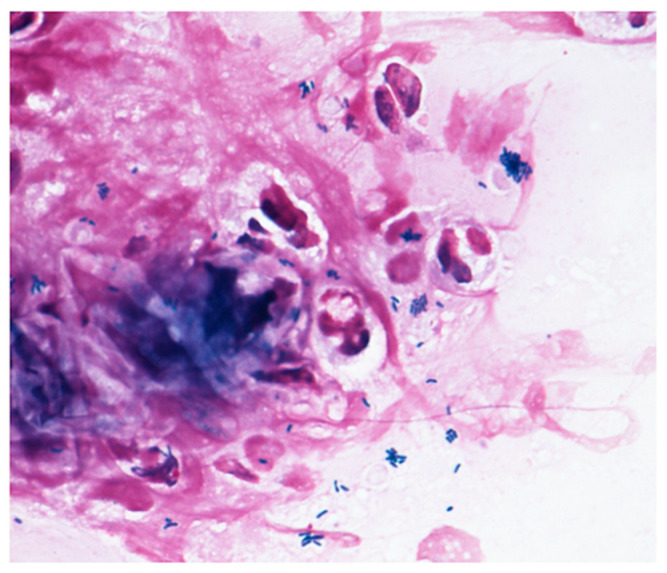
Gram staining. The Gram stain procedure revealed that gram-positive rod-shaped bacteria were phagocytosed by polymorphonuclear leucocyte.

**Table 1 microorganisms-09-00254-t001:** Representative cases with ocular surface infection caused by Corynebacterium species.

Authors (Ref. Number)	Year	Country	Age	Disease	Type of Strains	Lipophilic or Non-Lipophilic	Past History	Antibiotic Susceptibilities of Corynebacterium Species (Minimum Inhibitory Concentration; (μg/mL))							
								PCs	EM	CEPs	LVFX	CPFX	GM/TOB	VCM	CP
Badenoch PR et al. [14]	2016	Australia	94	Suture-related keratitis	*C. propinquum*	Non-lipophilic	Diabetes, Corneal transplantation	0.02	>256 (CLDM)	-	-	0.38	-	0.5	-
Duignan ES et al. [15]	2016	Ireland	52	Keratitis	*C. pseudodiphtheriticum*	Non-lipophilic	Corneal inlay	-	-	-	S (MFLX)	-	-	-	S
Todokoro D et al. [16]	2015	Japan	44	Contact-related keratitis	*C. propinquum*	Non-lipophilic	Contact lens wear, Proliferative diabetic retinopathy	-	>256	0.125	1	2	0.5	1	-
Ruoff KL et al. [17]	2010	USA	84	Keratitis	*C. macginleyi*	Lipophilic	Fuchs’ endothelial dystrophy	0.016	-	-	-	0.032	0.064	0.5	-
Alsuwaidi AR et al. [18]	2010	Canada	54	Conjunctivitis	*C. macginleyi*	Lipophilic	Health care worker	S	R	S	S	S	S	S	S
Inata K et al. [27]	2009	Japan	23	Contact-related keratitis	*C. mastidis*	Lipophilic	Contact lens wear	-	<0.016	0.125	0.064	0.125	<0.064	0.5	4
Eguchi et al. [8]	2008	Japan	58	Conjunctivitis	*C. macginleyi*	Lipophilic	-	-	-	-	>32	>32	-	-	-
Eguchi et al. [8]	2008	Japan	72	Conjunctivitis	*C. macginleyi*	Lipophilic	-	-	-	-	>32	>32	-	-	-
Eguchi et al. [8]	2008	Japan	58	Conjunctivitis	*C. macginleyi*	Lipophilic	-	-	-	-	>32	>32	-	-	-
Eguchi et al. [8]	2008	Japan	78	Conjunctivitis	*C. macginleyi*	Lipophilic	-	-	-	-	>32	>32	-	-	-
Suzuki T et al. [19]	2007	Japan	74	Suture-related keratitis	*C. macginleyi*	Lipophilic	Corneal transplantation	16	2	0.5	>128	128	<0.13	0.5	-
Suzuki T et al. [19]	2007	Japan	49	Suture-related keratitis	*C. macginleyi*	Lipophilic	Corneal transplantation	16	<0.13	0.25	64	8	<0.13	0.5	-
Giammanco G. M et al. [20]	2002	Italy	65	Corneal ulcers	*C. macginleyi*	Lipophilic	Trauma	S	S	S	-	S (ENX)	S	S	S
Li A et al. [21]	2000	China	86	Keratitis	*C. pseudodiphtheriticum*	Non-lipophilic	Pneumonia, Trichiasis, Lagophthalmos	R	-	R *	-	-	-	S	R
Heidemann DG et al. [25]	1991	USA	80	Keratitis	*C. striatum*	Non-lipophilic	Proliferative diabetic retinopathy	S	S	S *	-	-	-	S	-
Rubinfeld RS et al. [26]	1989	USA	81	Keratitis	*C. striatum*	Non-lipophilic	Contact lens wear, Aphakia	S	S	S *	-	-	-	S	-
Rubinfeld RS et al. [26]	1989	USA	11	Suture related keratitis	*C. xerosis*	Non-lipophilic	Post corneal laceration	-	S	S *	-	-	S	S	S
Funke et al. [7]	1998	Switzerland	47 ^a^	Corneal ulcer (*n* = 3), Conjunctivitis (*n* = 12)	*C. macginleyi*	Lipophilic	Contact lens wear, Eyelid closure problems	<0.01–0.125 ^b^	<0.03–>64 ^b^	0.5–16 ^b^	0.125–1 (Ofloxacin)^b^	0.06–0.125 ^b^	<0.06–0.5 ^b^	0.5–1 ^b^	2–4 ^b^

S: Sensitive, R: Resistant, *: First-generation cephalosporins, M: Male, F: Female, a: Average age, b: Range of antibiotic susceptibility in 15 patients, PCs: penicillins, EM: Erythromycin, CEPs: Cepharosporins, LVFX: Levofloxacin, CPFX: Ciprofloxacin, GM: Gentamicin, TOB: Tobramycin, VCM: Vancomycin, CP: Chloramphenicol, MFLX: Moxifloxacin, ENX: Enoxacin, CLDM: Clindamycin.

**Table 2 microorganisms-09-00254-t002:** Antibiotic susceptibilities of Corynebacterium species in clinical research.

Authors (Ref. Number)	Year	Country	Samples	No. of Strains	Type of Strains	% of Susceptibilities to Antibiotics in *Corynebacterium* Species							
						PCs	EM	CEPs	LVFX	CPFX	GM	VCM	CP
Hoshi et al. [23]	2020	Japan	conjunctiva (*n* = 46), nose (*n* = 4)	50	-	100	100	-	64	-	98 (TOB)	-	-
Deguchi et al. [22]	2018	Japan	conjunctiva	77	-	-	28	100	0	-	-	100	88
Watson et al. [60]	2016	Australia	cornea	8	-	-	-	60	-	86	-	100	75
Das et al. [9]	2015	India	cornea	22	*C. propinquum* (*n* = 3), *C. pseudodiphtheriticum* (*n* = 1), *C. striatum* (*n* = 1), *C. bovis* (*n* = 1)	-	-	80	-	50	-	89	56
Eguchi et al. [32]	2013	Japan	ocular surface	20	-	-	45	100	25	25	95	100	55
Bharathi et al. [59]	2010	India	ocular surface, vitreous humor	207	-	-	-	91	-	62	49	90	49
Eguchi et al. [8]	2008	Japan	ocular surface	21	*C. macginleyi* (*n* = 16), *C. mastitidis* (*n* = 4), *C. accolens* (*n* = 1)	-	-	-	43	43	-	-	-
Cameron et al. [57]	2006	Australia	cornea	8	*C. macginleyi* (*n* = 4)	-	-	88	-	100	75	-	88
Joussen et al. [28]	2000	Germany	conjunctiva	10	*C. macginleyi* (*n* = 10)	100	70	-	80	-	100	-	67

PCs: penecillins, EM: Erythromycin, CEPs: Cepharosporins, LVFX: Levofloxacin, CPFX: Ciprofloxacin, GM: Gentamicin, TOB: Tobramycin, VCM: Vancomycin, CP: Chloramphenicol, OFLX: Ofloxacin.

**Table 3 microorganisms-09-00254-t003:** Antibiotic susceptibilities of *Corynebacterium* species from the conjunctiva and nose.

Samples	No. of Strains	% of Susceptibilities to Antibiotics in *Corynebacterium* Species						
		EM	CMX	LVFX	ABK	FOM	VCM	CP
Conjunctiva	77	40	96	45	95	1	100	73
Nose	176	69	94	74	94	3	100	72
Total	253	40	95	65	94	3	100	72

EM: Erythromycin, CMX: Cefmenoxime, LVFX: Levofloxacin, ABK: Arbekacin, FOM: Fosfomycin, VCM: Vancomycin, CP: Chloramphenicol.

## Data Availability

Not applicable.

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
