# Peer review of "Current Evidence for Corynebacterium on the Ocular Surface"

_microorganisms, 2021, doi:10.3390/microorganisms9020254_

Round 1

Reviewer 1 Report

In the present manuscript, authors have reviewed the existence of several species of Corynebacterium on the ocular surfaces, their antimicrobial susceptibilities, molecular mechanisms of resistance, and risks to patients having several health issues. Overall, the manuscript is well written, informative and paragraphs are cohesive. I did not find any major issue with the manuscript.

Minor issue:

  1. Lines 29-31: please cite the relevant article.
  2. Table 1. Some texts are not aligned.

Reviewer 2 Report

Partition analysis of the revision:

Corynebacterium species are commonly found in the conjunctiva of healthy adults and are recognized as non-pathogenic bacteria. In recent years, however, Corynebacterium species have been reported to be potentially pathogenic in various tissues. We studied Corynebacterium species on the ocular surface and examined various Corynebacterium species in terms of antimicrobial susceptibility and underlying molecular resistance mechanisms.

I agree with the evidence that the use of third generation cephems should be a reasonable and pragmatic approach for treating eye infections caused by Corynebacterium species.

  • Introduction: review the syntax discussed and deal directly with the topic in question and not be too superficial.
  • - Methods: Correctly analyzed and clear here no changes are needed
  • - In the discussion the use of new eye drops in the treatment of this infection that is creating more and more problems should be studied in depth, so I recommend the bibliographic entries to use and insert: PMID: 32452982; PMID: 30950680; PMID: 28796877
  • - A revision of the grammar and of the English language is required, in particular some syntax of the introduction, however, to revise the entire text for grammatical uniformity.
  • These suggestions are necessary on my part for acceptance
  • I believe that the paper should be well revised following a major revision for the purpose of subsequent acceptance and is not to be rejected.

Round 2

Reviewer 2 Report

The corrections were made successfully.